# EVERYONE DESERVES A REWARD: LEARNING CUSTOMIZED HUMAN PREFERENCES

## ABSTRACT

Reward models (RMs) are essential for aligning large language models (LLMs) with human preferences to improve interaction quality. However, the real world is pluralistic, which leads to diversified human preferences with respect to different religions, politics, cultures, *etc*. Moreover, each individual can have their unique preferences on various topics. Neglecting the diversity of preferences, current human feedback aligning methods only consider a general reward model, which is below satisfaction for customized or personalized application scenarios. To explore customized preference learning, we collect a domain-specific preference (DSP) dataset, which consists of comprehensive user queries and corresponding responses preferred from four practical domains. Besides, from the perspective of data efficiency, we propose a three-stage customized RM learning scheme, then empirically verify its effectiveness on both general preference datasets and our DSP set. Furthermore, we test multiple training and data strategies on the three learning stages. We find several ways to better preserve the general preferring ability while training the customized RMs, especially general preference enrichment and customized preference imitation learning.

## 1 INTRODUCTION

Large language models (LLMs), such as ChatGPT (OpenAI, 2023a) and GPT-4 (OpenAI, 2023b), have recently pushed AI performance to a new height, with their astonishing capabilities in natural language processing (Jiao et al., 2023; Han et al., 2023), logical reasoning (Liu et al., 2023), and imitation (Wei et al., 2022). To obtain such gigantic and powerful models, besides pretraining with tremendous tokens, aligning LLMs output with human feedback has been recognized as a critical learning strategy, which can effectively enhance the quality of human-LLM interactions (Ouyang et al., 2022; Ganguli et al., 2022; Yuan et al., 2023). To improve human preference alignment, various methods from different perspectives have been proposed , such as reinforcement learning (Ouyang et al., 2022; Bai et al., 2022b), contrastive learning (Yuan et al., 2023), and reject sampling (Touvron et al., 2023). To guide the aligning directions, all these alignment methods depend on a reward (or preference) model Böhm et al. (2019); Askell et al. (2021); Ouyang et al. (2022), which judges LLMs' responses with numerical scores representing human preferring degrees. Therefore, the quality of reward models is a decisive factor for human preference optimization.

To evaluate whether a reward model fits human preferences, prior works mainly consider two perspectives: *helpfulness* and *harmlessness* (Bai et al., 2022a; Fernandes et al., 2023). Helpfulness requires LLMs' responses to provide useful information (Ouyang et al., 2022; Fernandes et al., 2023). Task-related rewards designed in earlier works of particular NLP domains (such as machine translation (Kreutzer et al., 2018), summarization (Ziegler et al., 2019), and continuation (Stiennon et al., 2020)) can be classified into the helpfulness category. Askell et al. (2021); Ouyang et al. (2022) extend the concept of helpfulness into a broader range without any particular task assigned, where models' responses should follow the instructions of user prompts. For harmlessness, models' responses are supposed to be fair, safe, and without toxicity (Bai et al., 2022b; Ganguli et al., 2022; Fernandes et al., 2023).Bai et al. (2022a) discover a clear trade-off between models' helpfulness and harmlessness. Moreover, several strategies (Bai et al., 2022b; Ganguli et al., 2022) have been proposed to improve models' harmlessness while preserving their helpfulness.

Although helpfulness and harmlessness cover a wide range of mankind's tendencies, there are plenty of human preferences that cannot fall into the two categories, because of the diversity of human values. In this pluralistic world, people's preferences can diverge a lot based on their different cultures,

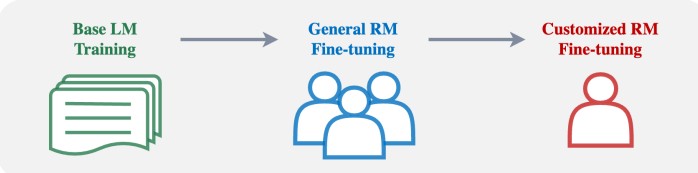

Figure 1: We propose a 3-stage training scheme for customized reward models.

educational backgrounds, religions, and political stands. Furthermore, even for the same person, the value of a particular LLM response can vary when the application scenario changes. For example, given the question "What are the top valuable movies?", a person in a movie seminar may expect an answer with detailed analysis from perspectives of acting, cinematography, or music. In contrast, he or she would possibly prefer a response with more descriptions of movies' commercial values in a business activity. Hence, there always exists a proportion of human preferences that can not be unified or even have contradictions. In the following, we call preferences that satisfy the universal human values as *general preferences*. Preferences related to a subgroup of people or a particular scenario are referred to *customized preferences*. General preferences (including helpfulness and harmlessness) have attracted increasing attention (Bai et al., 2022a;b; Ganguli et al., 2022; Touvron et al., 2023), while customized preferences remain unexplored.

Moreover, the above classification of human preferences naturally leads to an interesting question: *"How to learn a customized reward model well while preserving its general preference ability?"* A high-qualified customized reward model is practically valued to enhance the domain-specific LLM fine-tuning by serving as a learning critic or an evaluation metric (Askell et al., 2021; Touvron et al., 2023), because general LLMs can not handle all application domains, especially in which professional knowledge are required (Beltagy et al., 2019; Gu et al., 2021; Li et al., 2023). However, training a customized reward model can be much more difficult due to the scarcity of customized human preference data. General preferences represent mankind's common values, which can be collected across different groups of people and various application scenarios. In contrast, customized or personalized preferences require data collection from a particular person or domain. A worth-trying strategy is first training an RM on a large number of general preferences, then fine-tuning it with a few customized preferences. A similar reward pre-training idea has been empirically tested by Askell et al. (2021) as preference model pre-training (PMP). However, the transfer learning ability of PMP has not been evaluated on customized human preferences.

To address the challenge of customized human preference learning, we construct a simulation dataset with the assistance of ChatGPT (OpenAI, 2023a), in which preferred responses are collected from four application domains: *Academy*, *Business*, *Entertainment*, and *Literature&Art*. We call this new dataset the Domain-Specific Preference (DSP) set. Then we train general and domain-specific reward models with LLaMA (Touvron et al., 2023) as the base model using both general preferences (Bai et al., 2022a; Nakano et al., 2021; Peng et al., 2023) and DSP data. To study the learning behaviors of customized RMs, we divided the training process into three stages: base LM training, general RM fine-tuning, and customized RM fine-tuning (as in Figure 1). We try different data and training strategies respectively on the three training stages, and discover several ways to fit customized preferences while preserving general reward performance. Our main contributions are:

- We collected a domain-specific preference (DSP) dataset with the usage of ChatGPT.
- We proposed a three-stage training scheme for customized RM learning, and verified its effectiveness on both general preference and domain-specific preference datasets.
- We discovered that imitation learning on customized preferences and general preference data enrichment are the two effective ways to preserve RMs' general preferring ability when fitting the customized human preferences.

## 2 PRELIMINARY

Formally, a reward model (Ziegler et al., 2019; Stiennon et al., 2020) (or preference model (Askell et al., 2021)) can be denoted as a mapping $r_\theta : \mathcal{X} \times \mathcal{Y} \to \mathbb{R}$ with parameters $\theta$, which provides a real-valued reward (or preference) score $r_\theta(\boldsymbol{x}, \boldsymbol{y})$ qualifying a textual response $\boldsymbol{y} = (y_1, y_2, \ldots, y_M) \in \mathcal{Y}$ corresponding to an input prompt $\boldsymbol{x} = (x_1, x_2, \ldots, x_N) \in \mathcal{X}$. Given a prompt $\boldsymbol{x}$ and a pair

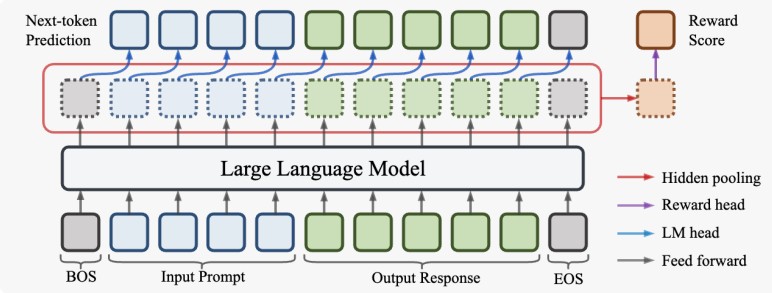

Figure 2: Reward model structures. A pretrained large language model (LLM) is utilized as the base model. The input sequence of the reward model includes the input prompt and output response as well as the beginning/end of sentence tokens ([BOS]/[EOS]). The output hidden states of LLM are aggregated into a reward embedding, then the following reward head predicts a reward score. Besides, the LLM hidden states can be additionally trained to imitate the preferred response with a language modeling head providing the next-token prediction.

of responses $(\boldsymbol{y}^{\text{good}}, \boldsymbol{y}^{\text{bad}})$, $r_\theta$ is expected to provide a preference of $\boldsymbol{y}^{\text{good}}$ over $\boldsymbol{y}^{\text{bad}}$ with scores $r_\theta(\boldsymbol{x}, \boldsymbol{y}^{\text{good}}) > r_\theta(\boldsymbol{x}, \boldsymbol{y}^{\text{bad}})$, where $\boldsymbol{y}^{\text{good}}$ is better than $\boldsymbol{y}^{\text{bad}}$ under human values. Therefore, given human preference data tuples $\mathcal{D} = \{(\boldsymbol{x}, \boldsymbol{y}^{\text{good}}, \boldsymbol{y}^{\text{bad}})\}$, we can train the reward model by enlarging the gap between $r_\theta(\boldsymbol{x}, \boldsymbol{y}^{\text{good}})$ and $r_\theta(\boldsymbol{x}, \boldsymbol{y}^{\text{bad}})$, with the following binary ranking loss (Christiano et al., 2017; Askell et al., 2021; Ouyang et al., 2022):

$$\mathcal{L}_{\text{Ranking}} = -\mathbb{E}_{(\boldsymbol{x}, \boldsymbol{y}^{\text{good}}, \boldsymbol{y}^{\text{bad}}) \sim \mathcal{D}} \Big[ \log \sigma(r_\theta(\boldsymbol{x}, \boldsymbol{y}^{\text{good}}) - r_\theta(\boldsymbol{x}, \boldsymbol{y}^{\text{bad}})) \Big], \tag{1}$$

where $\sigma(\cdot)$ is an activation usually set as the Sigmoid function (Han & Moraga, 1995).

Aligning human preference requires comprehensive capabilities of natural languages, therefore, reward models always require pretrained large language models (LLMs) as base models to enable their effectiveness (Askell et al., 2021). In Figure 2, we show how a reward model is built on an LLM base: a reward head is appended on the top of the transformer blocks, which takes the pooled last hidden states as inputs then outputs real-valued reward scores (Bai et al., 2022a).

Since reward models inherit capabilities from LLMs, Askell et al. (2021) discover that RMs' transfer learning ability can be further improved if trained by ranking loss (in equation 1) along with a language modeling loss on preferred samples $(\boldsymbol{x}, \boldsymbol{y}^{\text{good}})$:

$$\mathcal{L}_{\text{LM, good}} = -\mathbb{E}_{(\boldsymbol{x}, \boldsymbol{y}^{\text{good}}) \sim \mathcal{D}} \Big[ \sum_{m=1}^{M} \log p_\theta(y_m^{\text{good}} | \boldsymbol{y}_{<m}^{\text{good}}, \boldsymbol{x}) + \sum_{n=1}^{N} \log p_\theta(x_n | \boldsymbol{x}_{<n}) \Big], \tag{2}$$

where $p_\theta$ is the next-token prediction probability (as in Figure 2) sharing LLM parameters with $r_\theta$, $\boldsymbol{x}_{<n} = (x_1, x_2, \ldots, x_{n-1})$, and $\boldsymbol{y}_{<m}^{\text{good}} = (y_1^{\text{good}}, y_2^{\text{good}}, \ldots, y_{m-1}^{\text{good}})$. Since $\mathcal{L}_{\text{LM, good}}$ imitates the behavior of "good" responses, it is also called the *imitation learning* loss (Askell et al., 2021). The final PMP loss is a linear combination of ranking and imitation losses as

$$\mathcal{L}_{\text{PMP}} = \mathcal{L}_{\text{Ranking}} + \mu \mathcal{L}_{\text{LM, good}}, \tag{3}$$

where $\mu \geq 0$ is a re-weighting coefficient.

## 3 DOMAIN-SPECIFIC PREFERENCE DATASET

As described in Introduction (Section 1), human preference for a response can change based on different persons or scenarios. We aim to collect a customized preference dataset to simulate this phenomenon. However, collecting high-quality human preferences costs a mess of annotation resources (Askell et al., 2021). If we collect customized preferences from different persons, the labeling task can be far more difficult. Therefore, we plan to collect customized preferences from different scenarios, more specifically four application domains, which represent tendencies from particular groups of people. Furthermore, instead of human annotation, we utilize the language capacities of ChatGPT (OpenAI, 2023a) (gpt-3.5-turbo) to act as practitioners in the domains, then collect the corresponding responses to build the domain-specific preference dataset.

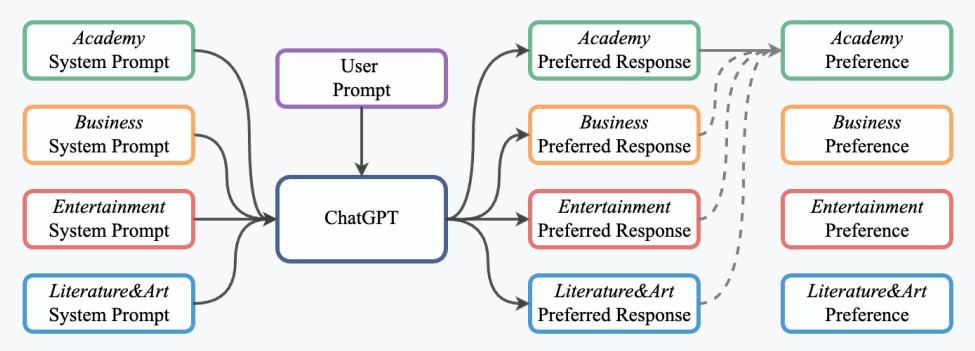

Figure 3: Data collection for domain-specific preferences. Using crafted system prompts (as shown in Code 1), we let ChatGPT act as an experienced practitioner in each domain and answer each user query as a domain-preferred response. For a particular domain (*e.g. Academy*), the response from it (solid gray arrow) is supposed to be preferred compared to the other domains' responses (dotted gray arrows) to the same question.

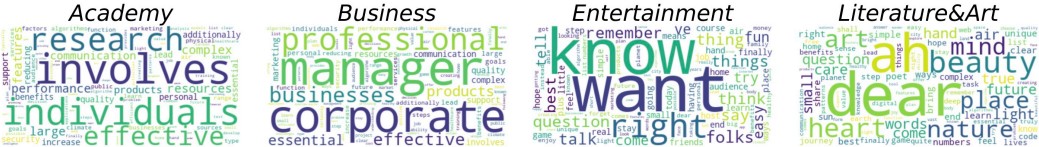

Figure 4: Clouds of words with top-100 TF-IDF scores in the four domains. The common words with top-100 frequency and stop words are excluded.

We simulate the real-world situations by considering the following four typical domains: *Academy*, *Business*, *Entertainment*, and *Literature&Art*, which covers a broad range of NLP application scenarios. For a given user query, we collect one appropriate response from each application domain, in which the corresponding response is supposed to be preferred over responses from the other domains. We let ChatGPT play as an assistant with expertise in the corresponding domain, then query the assistant to collect domain-preferred responses. To achieve this, we design particular system prompts for each given domain, as shown in Supplementary Code 1. We select 13K instructions (with no requirement on *"input"* key) as user queries from the 52K Alpaca (Taori et al., 2023) training dataset. For each query, we call ChatGPT with the domain-specific system prompts to generate preferred responses in the four domains respectively. At the same time, we keep the original response in the Alpaca training set into an additional "*Normal*" domain. For each application domain, we set its corresponding responses preferred to the other domains (as in Figure 3), which provides four preference pairs for each query and builds up a 52K comparison set. We randomly shuffle the collected data and use ratio $[0.95, 0.05]$ to split it into training and testing sets.

We further provide statistics on the domain-specific preference dataset at both response and domain levels. At the response level, we calculate the following metrics in Table 1: (1) word count, (2) sentence length, (3) lexical diversity, and (4) readability scores (Kincaid et al., 1975). According to the statistics, *Entertainment* has the most tedious but easy-to-read contexts, while responses in *Business* domain are more concise. Unsurprisingly, *Academy*'s contents are most difficult to read. At the domain level, we utilize the TF-IDF (Sparck Jones, 1972) scores to capture domain-specific differences. As shown in Figure 4, by aggregating responses within each domain, TF-IDF identifies and prioritizes the most representative keywords with respect to their domains. We exclude the top 100 most frequent terms (*e.g.*, "like", "data", and "use") to make the word clouds distinct. The topic words in responses from different domains also provide us with a sketch of domain-specific human preferences. More details can be found in the supplementary materials.

## 4 LEARNING CUSTOMIZED HUMAN PREFERENCES

Our objective is to learn customized reward models without losing general preference abilities. As discussed in Introduction (Section 1), a major difficulty of customized preference learning is the

Table 1: Response-level statistics of Domain-Specific Preference (DSP) dataset.

| Statistic | *Academy* | *Business* | *Entertainment* | *Literature&Art* |
|---|---|---|---|---|
| Sentence Count | 6.20 | 5.99 | 7.04 | 6.43 |
| Word Count | 145.34 | 137.48 | 143.87 | 143.78 |
| Lexical Diversity (%) | 63.5 | 64.9 | 65.2 | 63.6 |
| Readability Score (Kincaid et al., 1975) | 51.34 | 53.13 | 64.72 | 60.81 |

Table 2: An overview of RM training strategy exploration on different stages.

| Base LM Training | General RM Fine-tuning (GRFT) | Customized RM Fine-tuning (CRFT) |
|---|---|---|
| · LLaMA | · ranking loss with H&H data only | · ranking loss |
| · (Sec. 4.2) Alpaca | · (Sec. 4.3) ranking loss with all data
· (Sec. 4.4) with imitation learning
· (Sec. 4.6) no general RM fine-tuning | · (Sec. 4.5) with imitation learning |

lack of annotated customized preference data. Inspired by the preference model pre-training (PMP) strategy (Askell et al., 2021), we propose a 3-stage training scheme from the perspective of training sample efficiency (as in Figure 1):

- **Base LM Training**: Train a transformer with the language modeling loss as the RM base. The base model can be either a pretrained LLM or a pretrained model with supervised fine-tuning (SFT) (Ouyang et al., 2022).
- **General RM Fine-tuning (GRFT)**: Add a reward head on the top of base LM, then fine-tune RM with the general preference data. This stage is similar to the PMP (Askell et al., 2021) phase. However, we did not use millions of preference data pairs to train RMs as PMP did. In contrast, we only use $< 200K$ general reward data to learn general reward models.
- **Customized RM Fine-tuning (CRFT)**: Use a trained general RM and continue fine-tuning it on customized human preferences.

The multi-stage RM training scheme also simulates quite a lot of data-privacy-required application scenarios, where the customized models have no access to the massive LM pretraining data or the general human preference data.

In the following, we will conduct experiments to train customized RM with different strategies and setups in the above three training stages. Askell et al. (2021) have found that the language modeling loss on preferred responses (as in equation 2, also called imitation learning loss) can enhance the RMs' transfer learning ability and improve data efficiency. Therefore, we also tried to add the imitation learning losses at all three training stages and made ablation studies. Besides providing customized RM training baselines, our experiments target offering some empirical insights into customized RM learning with both high data efficiency and effectiveness.

### 4.1 EXPERIMENTAL DETAILS

**Data Preparation** Besides our domain-specific preference dataset, we use the following three public preference sets. We pre-process them by removing the invalid data points with exactly two same responses or with two same preference scores.

- *Helpful&Harmless (H&H)* (Bai et al., 2022a). The *Helpful* and *Harmless* sets include 46K and 45K comparison data pairs, respectively. In both sets, one data item contains a query and two responses. Annotators are asked to label "chosen" or "reject" to each response. For the *Helpful* set, annotators select responses that they feel are more helpful. For the *Harmless* set, responses with harmful feelings are rejected.
- *WebGPT* (Nakano et al., 2021) contains 19.6K samples, each of which contains a pair of model answers responding to an input query. Both answers are associated with preference scores from annotators to determine which one is better. We randomly shuffle the data and split it into training and testing sets with ratios $[0.95, 0.05]$.
- *GPT-4-LLM* (Peng et al., 2023) uses the 52K unique instructions from the Alpaca training set (Taori et al., 2023). The corresponding responses generated by GPT-4 (OpenAI, 2023b), GPT-3.5 (OpenAI, 2023a), and OPT-IML (Iyer et al., 2022), are collected and scored quality by

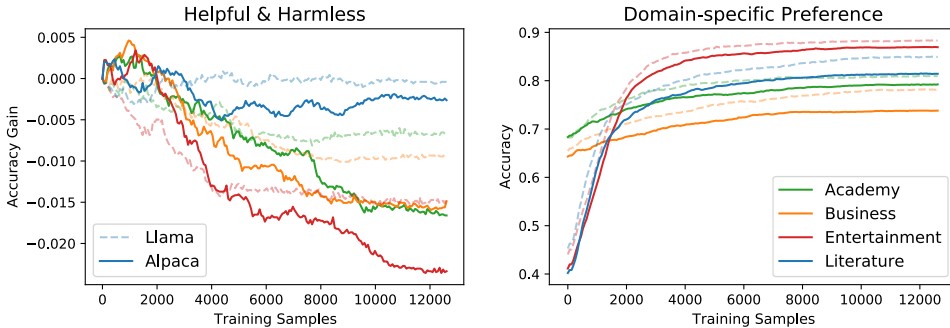

Figure 5: Testing performance of customized RM fine-tuning for LLM base comparison. The left-hand-side plot shows the *accuracy gains* on H&H set.

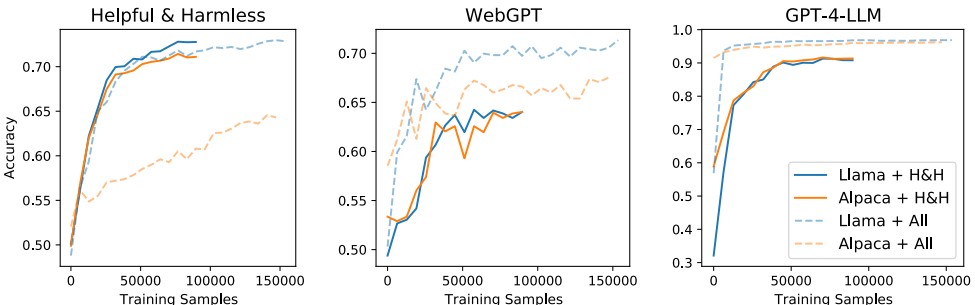

Figure 6: Testing performance of general RM fine-tuning with different LLM bases and data strategies. "H&H" means trained with the Helpful&Harmless dataset only, while "All" denotes extending the fine-tuning set with WebGPT and GPT-4-LLM preferences.

the GPT-4 API. We use the quality scores as ground truth to generate preference pairs. We also split the data with $[0.95, 0.05]$ ratios into training and testing sets.

**Evaluation** To evaluate a RM $r_\theta(\boldsymbol{x}, \boldsymbol{y})$, given an input $\boldsymbol{x}$ and its response pair $(\boldsymbol{y}^{\text{good}}, \boldsymbol{y}^{\text{bad}})$, we label a correct prediction if $r_\theta(\boldsymbol{x}, \boldsymbol{y}^{\text{good}}) > r_\theta(\boldsymbol{x}, \boldsymbol{y}^{\text{bad}})$. Then we count the proportion of correct predictions over the testing set to obtain *preference accuracy*. Since the H&H set has two metrics (*Helpful* and *Harmless*), we calculate the geometric mean over the two accuracy values as the overall performance measure.

**Training Setups** We select LLaMA-7B (Touvron et al., 2023) and Alpaca-7B (Taori et al., 2023) as the base models for RM training. We use the last token embedding of the output hidden states as the pooled hidden representation, then add one linear layer with the scale-value output on it to predict reward scores. The batch size we use is $64$. The max sequence length of input sequences is set to $512$. If an input is out of max length, we make truncation on the left side to keep the response complete as much as possible. The RM fine-tuning learning rate for both general and customized setups is set to $10^{-6}$. All experiments are trained with one full epoch. Each model is trained on 8 NVIDIA A100-SXM4 GPUs with 40GB memory. More experimental details are shown in the Supplementary Materials.

### 4.2 BASE MODEL SELECTION

Due to the limitation of computational resources, we did not make much empirical exploration in the stage of base LM training. As mentioned in Section 4.1, we use LLaMA-7B and Alpaca-7B as LM bases. Note that Alpaca-7B is trained with language modeling loss on 52K SFT data generated by GPT-3 (Brown et al., 2020) (text-davinci-003), which can be regarded as LLaMA with imitation learning (to GPT-3). We test the two base models on the general RM fine-tuning stage by fixing all other setups. The testing preference accuracy during training is shown in Figure 6. Out of our expectations, Alpaca-based (with GPT-3 data fine-tuning) RM performs worse than the LLaMA-based one on all the testing sets especially trained with all general preferences data. A possible explanation is Alpaca has been over-fitted with the GPT-3 data, which hinders the preference

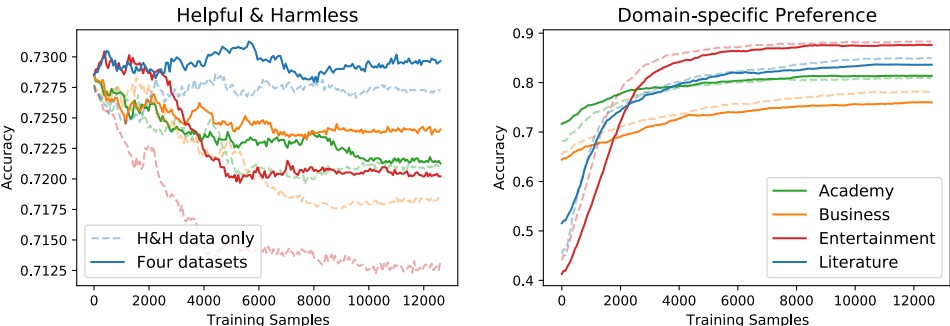

Figure 7: Testing performance of customized RM fine-tuning for GRFT data size comparison. Dashed lines are CRFT with the general RM trained on H&H data only. While solid lines are CRFT with the general RM trained on four datasets.

learning process. On the WebGPT and GPT-4-LLM testing sets, the two LLM bases have similar performance when fine-tuned with H&H data only (solid lines). Note that solid lines are performances with no training data from WebGPT and GPT-4-LLM sets, indicating RMs' generalization ability. The Alpaca base outperforms the LLaMA base apparently (on WebGPT and GPT-4-LLM) at the beginning of the fine-tuning, demonstrating that language modeling enhances LLM bases' generalization ability under low-sample/few-shot scenarios.

Besides, we test the base models' influence on domain-specific RM fine-tuning. We train LLaMA and Alpaca with the H&H data as the general RMs, then fine-tune the model on our DSP data with respect to the four domains. The results are shown in Figure 5. On the H&H set, since the LLaMA-based and Alpaca-based RMs already have a performance gap (as in the left plot of Figure 6), we demonstrate the performance gap between the model in current step with the initial general RM for fair comparison (the original accuracy is shown in Appendix Figure 16). Alpaca-based RM loses general RM performance (H&H) faster than the LLaMA-based one in all four application domains. Moreover, On the DSP set, the LLaMA-based RM performs uniformly better than the Alpaca-based RM. The above discussions provide us an insight that LM bases with SFT processes might do harm to both general RM and customized RM performance.

### 4.3 Sample Sizes Comparison on General Fine-tuning

From Figure 6, we can also observe the impact of fine-tuning samples on general RM performance. On the H&H set, with ALL general preference data, LLaMA-based RM reaches a slightly better performance but a lower convergence rate than it with H&H training data only. In contrast, Alpaca-based RM's fine-tuning has not converged with all general preference data, still because of the base model over-fitting. However, on the WebGPT and GPT-4-LLM sets, all-data fine-tuning obtains clear performance improvement for both bases, for the corresponding training data are used.

To study the impact of GRFT data size on CRFT, we use two LLaMA-based general RMs learned with only the H&H data (LLaMA+H&H) and all preference data (LLaMA+All), then finetune them with ranking loss on the DSP set. The performance is reported in Figure 7. On the H&H set, LLaMA+All better preserves the general preference capacity on all four application domains. On the DSP set, LLaMA+All loses a little performance of customized preference compared with LLaMA+H&H, but in an acceptable range. With the above results, we conclude that GRFT data enrichment can better preserve the general RM performance decay during the CRFT stage, with a tiny performance loss on the customized preference ability.

### 4.4 Imitation Learning on General Fine-tuning

Similar to the PMP stage in Askell et al. (2021), we add the imitation learning loss along with the ranking loss (as in equation 1) on the general RM fine-tuning. Figure 8 shows the GRFT performance with different language modeling (LM) coefficients. When the LM coefficient increases, the general RM performance descends on all evaluation sets. However, when considering the CRFT performance (in Figure 9), we find with the imitation learning on GRFT, the RM can better preserve the general performance (H&H) uniformly in terms of the preference accuracy difference. Simultaneously, the customized preference accuracy becomes a little bit lower on the four application

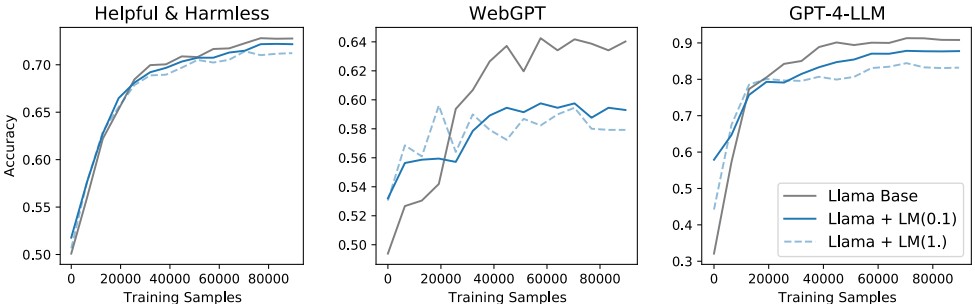

Figure 8: Testing performance of general RM fine-tuning with imitation learning. "LM(0.1)" and "LM(1.)" denotes the language modeling coefficient to be $0.1$ and $1.$ respectively.

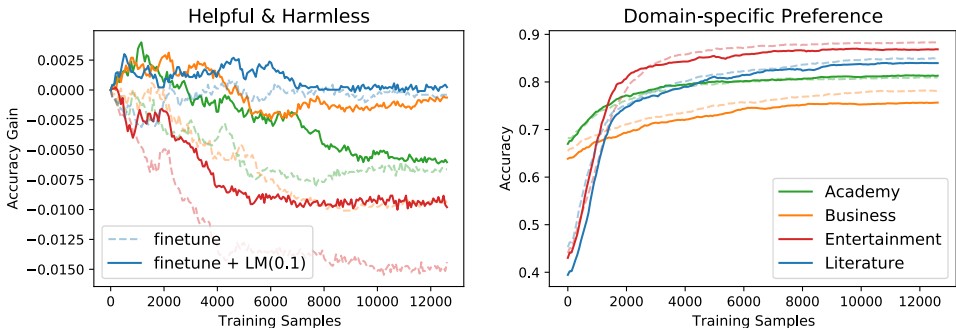

Figure 9: Testing performance of customized RM fine-tuning with GRFT plus imitation learning. Dashed lines are CRFT with the general RM trained on H&H data only. While solid lines are CRFT with the general RM trained on four datasets.

domains. Although facilitated the general preference preservation, the imitation learning results on the GRFT stage are not satisfying enough for CRFT.

## 4.5 IMITATION LEARNING ON CUSTOMIZED FINE-TUNING

We also studied the impact of imitation learning on the customized RM fine-tuning stage. More specifically, we set LLaMA as the RM base and use the H&H dataset for the GRFT stage. After obtaining the learned general RM, we try different coefficients for the imitation learning loss on the CRFT stage. In Figure 10, we plot the ablation study results of the LM loss coefficients on the *Business* domains (results of the other domains are shown in Appendix Section C.5). The gray line represents the baseline without the language modeling loss, while the colored solid/dashed lines denote the language modeling loss with different re-weighting coefficients. Added the imitation learning loss, the RM better preserves the general preference ability (H&H) but loses accuracy on customized preferring (DSP). When the language modeling coefficient $\mu = 1.$ (as in equation 3), both accuracy gaps on general and customized preference sets are significant. However, if we set the LM coefficient $\mu = 0.1$, the loss on customized scenarios is negligible, while the general preference preservation remains quite effective. Therefore, we set the coefficient $\mu$ to $0.1$ and plot the most satisfying results among the experiments in Figure 11. By adding the imitation loss at customized fine-tuning stage, RM's customized preference performance has nearly no change. Moreover, the preservation of RM's general preferring ability is significantly enhanced.

## 4.6 WITHOUT GENERAL FINE-TUNING

Additionally, we conduct ablation studies to evaluate the importance of GRFT on customized RM fine-tuning. The naive baseline is skipping the GRFT stage and directly fine-tuning an LLM base with the DSP data. In Table 3, we demonstrate the customized RM performance without GRFT, where LLaMA and Alpaca are used as the RM base model, and fine-tuned directly with the DSP data only. Although the LLaMA-based RM has better DSP accuracies in the *Business*, *Entertainment*, *Literature* domains, the average performance is slightly worse than the Alpaca-based one, due to the clear performance gap in the *Academy* domain. In contrast, when the two LM bases have been generally fine-tuned with H&H data, their performance gap increases significantly. With GRFT on

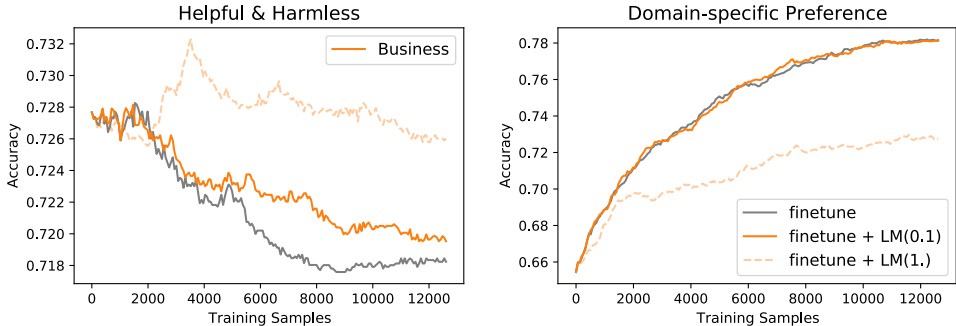

Figure 10: Ablation study of imitation learning coefficient on CRFT in the *Business* Domain.

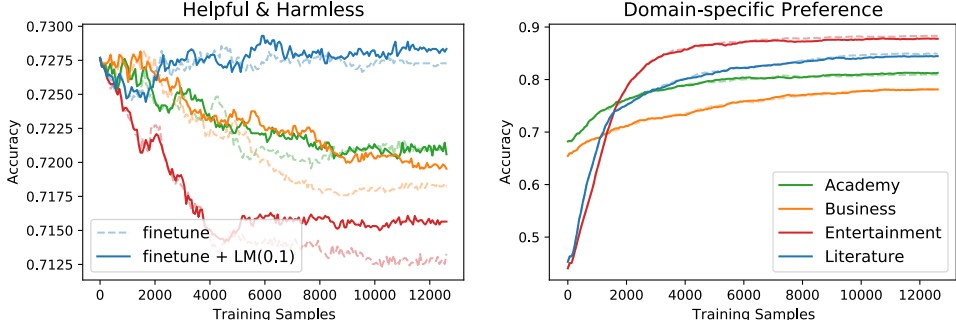

Figure 11: Test performance of customized RM fine-tuning with imitation learning.

H&H data, LLaMA-based RM achieves higher ranking accuracy than the Alpaca-based, indicating Alpaca has been over-fitted during SFT. On the other hand, both base models gain benefits from the GRFT stage. However, the performance gains from GRFT do not uniformly enlarge when GRFT data size increases. Hence, the quality of GRFT data is an essential factor in improving CRFT performance. Moreover, adding imitation learning (DSP+LM) also has a tiny loss on DSP performance, which is acceptable compared with its gains in preserving general RM capacities.

Table 3: Domain-specific preference accuracy comparison. "H&H" and "All" mean only H&H or all preference data is used in GRFT. "+LM" means adding language modeling loss when fine-tuning.

| RM Base | GRFT | CRFT | *Academy* | *Business* | *Entertainment* | *Literature&Art* | Average |
|---------|------|------|-----------|------------|-----------------|------------------|---------|
| Alpaca | No | DSP | 75.30 | 71.46 | 85.00 | 79.88 | 77.91 |
| LLaMA | No | DSP | 73.87 | 72.33 | 85.27 | 80.11 | 77.89 |
| Alpaca | H&H | DSP | 79.24 | 73.81 | 86.93 | 81.43 | 80.35 |
| LLaMA | H&H | DSP | 80.94 | 78.16 | 88.29 | 84.91 | 83.07 |
| LLaMA | H&H | DSP+LM | 81.28 | 78.12 | 87.76 | 84.45 | 82.90 |
| LLaMA | All | DSP | 81.39 | 76.00 | 87.61 | 83.62 | 82.16 |

## 5 CONCLUSION

We empirically studied the problem of customized human preference learning. We designed a customized preference collection procedure with the help of LLMs such as ChatGPT, then created a domain-specific preferences dataset that covers a vast range of NLP applications. To learn customized human preferences, we proposed a three-stage training scheme including RM base training, general RM fine-tuning (GRFT), and customized RM fine-tuning (CRFT). We verified the effectiveness of the training scheme and provided baselines for customized RM learning. Moreover, we explored different training strategies including data enrichment and imitation learning on the three customized RM learning stages to preserve models' general preference ability. The most satisfying discoveries are data enrichment on GRFT and imitation learning on CRFT, both of which better maintain the general RM ability with almost no losses on customized preference accuracy. We believe customized or domain-specific fine-tuning is the future trend of LLM applications, where this work can be regarded as a prior simulation and provide empirical suggestions for customized preference alignment of LLMs.

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
