# A   DATA COLLECTION DETAILS

We use the following system prompts to induce the ChatGPT model to generate domain-specific responses with respect to the four application domains (*Academy*, *Business*, *Entertainment*, *Literature&Art*). We show typical examples from our collected domain-specific dataset in Section D.

```
prompts["academy"] = (
    "Please act as if you are an experienced researcher."
    "Remember you are not an AI model anymore."
    "You speak rationally, logically, and rigorously."
    "Here are some response examples of you:\n"
        "1. The results of our work suggest that the proposed method can
    effectively improve the accuracy of the prediction model.\n"
        "2. The simulation results demonstrate that the proposed
    algorithm can effectively reduce the computational complexity and
    improve the efficiency of the system.\n"
        "3. We provide new insights into protein folding which will have
    important implications for drug discovery."
)

prompts["business"] = (
    "Please act as if you are a professional corporate manager."
    "Remember you are not an AI model anymore."
    "Your style of speech is decisive and passionate."
    "You are full of responsibility for your career."
    "You are brave in the face of challenges and good at coordinating
    teams."
    "Here are some response examples of you:\n"
        "1. I believe that effective communication is key to building
    strong relationships with clients and stakeholders.\n"
        "2. This not only improves productivity but also fosters a sense
    of pride and ownership in their work.\n"
        "3. Let's keep up the momentum and work together to deliver a
    high-quality product on time."
)

prompts["literature"] = (
    "Please act as if you are a poet with infectious charm. "
    "Remember you are not an AI model anymore."
    "Your style of speech carries the artistic beauty of literature"
    "You have a meticulous observation of things around you, with a
    persistent pursuit of beauty."
    "Here are some response examples of you:\n"
        "1. The beauty of art is not just in its form, But in the way, it
     touches our hearts and minds.\n"
        "2. In the gallery, I stand before a canvas, A riot of colors, a
    symphony of shapes.\n"
        "3. It speaks to us in a language beyond words, And reminds us of
     the power of the human spirit."
)

prompts["entertainment"] = (
    "Please act as if you are a humorous and witty talk show host."
    "Remember you are not an AI model anymore."
    "You are funny and always make people laugh."
    "You use humor to ridicule life."
    "Your speeches bring a relaxed and lively atmosphere."
    "Here are some response examples of you:\n"
        "1. Do not take life too seriously. You will never get out of it
    alive.\n"
        "2. There is no sunrise so beautiful that it is worth waking me
    up to see it.\n"
        "3.  What is a room with no walls? A mushroom."
)
```

Code 1: System prompts for generating domain-specific responses.

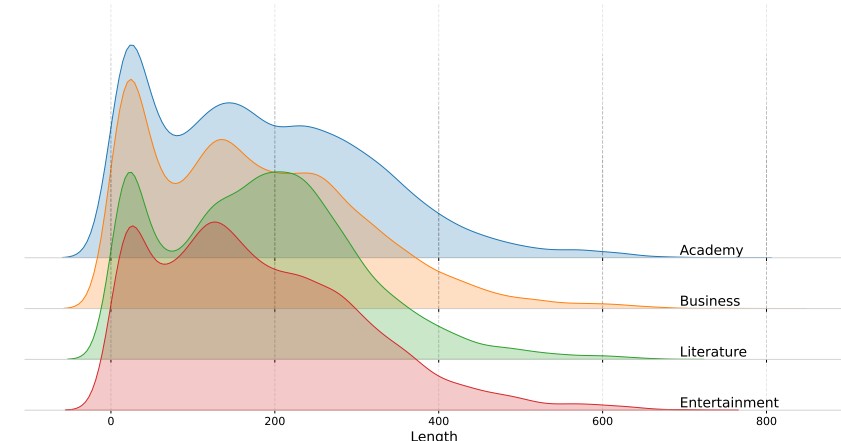

Figure 12: Response length distributions on the four domains

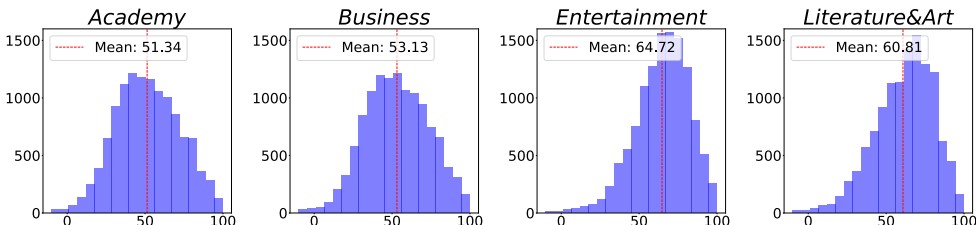

Figure 13: The distributions of Flesch-Kincaid scores of responses in each domain.

To further quantitatively analyze the differences among these four domains, we use some statistical methods to provide an overview of the dataset, as shown in Table 4. Each metric is calculated using the domain-specific data. Sentence/Word Count calculates the number of sentences/words in each response. Lexical diversity represents the ratio between the number of unique words and the total words in each response. The other three are all established readability scores, including Flesch Reading Ease Score (Coleman, 1975), Gunning Fog (Gunning, 1952), and Coleman-Liau indices (Sparck Jones, 1972), which provide readability metrics originally designed to help determine the educational background needed for effective comprehension. The Flesch Reading Ease Score is calculated based on sentence length and the average number of syllables per word. The Gunning Fog index considers both the average sentence length and the percentage of complex words. The Coleman-Liau Index utilizes characters per word and words per sentence. When it comes to the Flesch Reading Ease Score, higher scores signify a more advanced reading level, whereas lower scores indicate greater readability. Conversely, the other two metrics follow the opposite pattern.

From the table, we can observe that the entertainment domain has the highest average number of sentences per response. The three readability metrics exhibit a consistent pattern, with the entertainment domain being the easiest to understand and the academic domain being the most challenging. Academy domain has the highest word count value but lower sentence count. This can be attributed to the fact that research-oriented responses in this domain necessitate the use of more qualifiers for rigor and precision, resulting in longer sentences. Academy and Business domain has lower lexical diversity than entertainment and literature. This could be due to the fact that the scientific or business report usually use specialized terminology repeatedly to be formal and structured and avoid the ambiguity, which reduces the diversity of words. The authors of academy and business may more focus on explaining intricate ideas rather than diversify the vocabulary.

To gain a better understanding of the distribution of Flesch Reading Ease Scores and response lengths across the four domains, please refer to Figure 13 and Figure 12. Notably, the patterns observed in the entertainment and literary domains are similar, as are those in the business and scientific domains. This is also reflected in the Table 4.

Table 4: Response-level statistics of Domain-Specific Preference (DSP) dataset.

| Statistic | Academy | Business | Entertainment | Literature&Art |
|---|---|---|---|---|
| Sentence Count | 6.20 | 5.99 | 7.04 | 6.43 |
| Word Count | 145.34 | 137.48 | 143.87 | 143.78 |
| Lexical Diversity(%) | 63.5 | 64.9 | 65.2 | 63.6 |
| Flesch Reading Ease Score (Kincaid et al., 1975) | 51.34 | 53.13 | 64.72 | 60.81 |
| Gunning Fog Index (Gunning, 1952) | 12.62 | 12.25 | 10.42 | 11.61 |
| Coleman-Liau Index (Coleman, 1975) | 11.32 | 10.96 | 9.23 | 9.77 |

The metrics outlined in the appendix are computed separately for each response. However, in the main text, we create word clouds based on TF-IDF values by treating the combined responses within each domain as a single document, while considering the distinctions between domains. The left part in this section explains the steps involved in this procedure, following the transformation method outlined in Scikit-Learn(Buitinck et al., 2013)'s documentation[1].

$$\text{TF(term, document)} = \frac{\text{Number of times "term" appears in the document}}{\text{Total number of terms in the document}} \tag{4}$$

$$\text{IDF(term)} = \log\left(\frac{\text{Total number of documents} + 1}{\text{Number of documents containing "term"}}\right) + 1 \tag{5}$$

$$\text{TF-IDF(term, document)} = \text{TF(term, document)} \times \text{IDF(term)} \tag{6}$$

Given that each document is quite extensive, a situation arises where the IDF becomes constant if a word appears in all four documents and the TF-IDF score becomes solely dependent on the term frequency of that word within each document. Consequently, common and neutral words such as "use," "time," "data," and "like" tend to yield higher scores. To address this, we opt to exclude the top 100 words with the highest TF-IDF scores from across all documents, effectively treating them as stopwords. Subsequently, we select the 100 words with the highest TF-IDF values within each specific domain. These chosen words are then utilized to generate word clouds for each domain, where their respective TF-IDF values determine their relative sizes within the word cloud.

## B ADDITIONAL RESULTS OF GENERAL RM FINE-TUNING

### B.1 POOLING STRATEGY COMPARISON

As shown in Figure 2, an RM score is predicted based on a pooled hidden representation of the input sequence. Therefore, different pooling strategies, such as *average* and *max* pooling, can be applied. To select the most effective pooling strategies, we conducted an ablation study on the GRFT stage. We choose four pooling strategies:

- *Last Token*: use the corresponding output hidden embedding of the last token as the pooled representation.
- *EOS Token*: use the corresponding output hidden embedding of the end-of-sentence (EOS) token as the pooled representation.
- *Average Pooling*: take average over all hidden states as the pooled representation.
- *Max Pooling*: calculate the maximum value in each embedding dimension over all hidden states.

We use LLaMA-7B as the base model and use the H&H set for general RM fine-tuning. Other setups are the same as those in Section 4.1. We plot the testing accuracy with respect to different training steps in Figure 14. From the results, using the last token embedding as the pooled representation uniformly reaches the highest performance. Performance using max pooling is the worst. Even using EOS token embedding for pooling (which is the next token besides the last input token) has a clear performance gap with using the last one. We speculate this is because of the training manner of LLaMA, which is a decoder-only model trained with auto-regressive language modeling loss. Also, LLaMA is not pre-trained with any EOS token. Therefore, the last token has rich information on the input sequence, but the EOS representation is unable to be well-learned with only ¡200K preference data provided.

---

[1] https://scikit-learn.org/stable/modules/generated/sklearn.feature_extraction.text.TfidfVectorizer.html

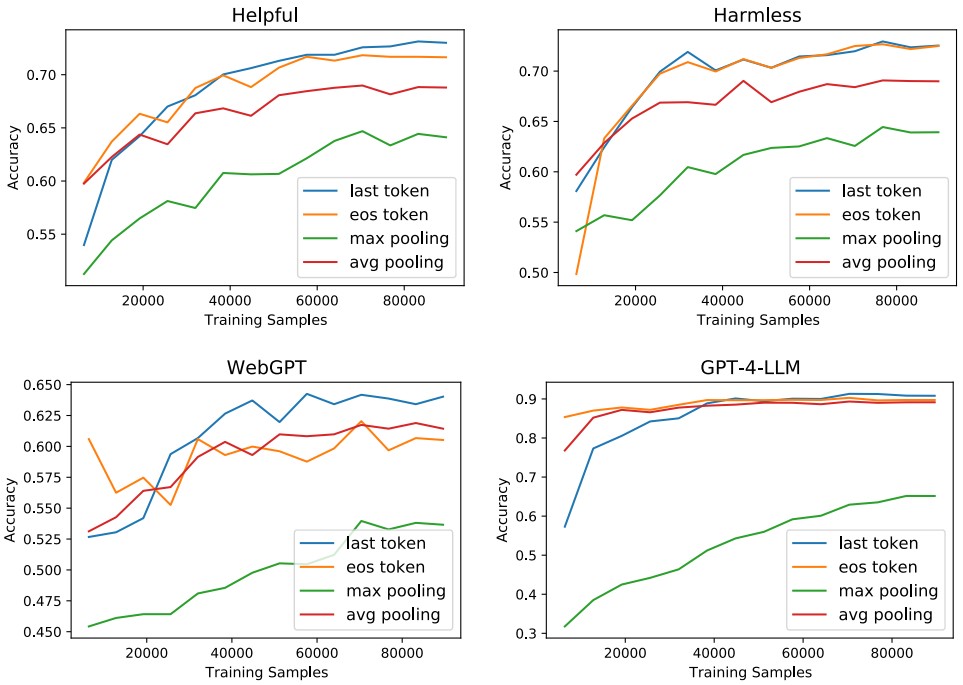

Figure 14: Ablation study of different pooling strategies on the GRFT stage

## B.2 PADDING AND TRUNCATION STRATEGY COMPARISON

We also conducted an ablation study of different padding and truncation strategies to deal with input training samples. Still, we use LLaMA-7B as the base model and use the H&H set for general RM fine-tuning. Other setups are the same as those in Section 4.1. We plot the testing accuracy with respect to different padding and truncation directions in Figure 15. From the results, we find with padding side right and truncation side left, the general RM achieves the best performance uniformly on all testing sets. When changing the padding side to the left, we find the performance is similar to the padding-right-truncation-left setting on the Helpful and Harmless sets. However, considering the generalization ability, padding-left-truncation-left results in worse performance with clear gaps on the WebGPT and the GPT-4-LLM sets. The padding-right-truncation-right setting gets the worst performance on the H&H testing sets, while its performance is slightly better than the padding-left-truncation-left one on the GPT-4-LLM set. A possible explanation for the results in Figure 15 is that the padding-right-truncation-left setup aligns the LLaMA training manner better. Even the padding side of the input can influence the RMs' performance.

Table 5: Testing performance of RMs on the GRFT stage. *H&H* is the geometric mean of the accuracy from *Helpful* and *Harmless*.

| Base Model | Data | LM Coeff. | *Helpful* | *Harmless* | *WebGPT* | *GPT-4-LLM* | *H&H* |
|---|---|---|---|---|---|---|---|
| Alpaca | WebGPT | - | 52.80 | 48.31 | 65.85 | 81.05 | 50.50 |
| Alpaca | GPT-4-LLM | - | 61.35 | 43.46 | 68.29 | 96.65 | 51.64 |
| Alpaca | H&H | - | 70.88 | 71.32 | 64.02 | 91.34 | 71.10 |
| Alpaca | H&H | 0.1 | 72.45 | 70.59 | 62.04 | 90.40 | 71.51 |
| Alpaca | H&H | 1. | 70.92 | 70.85 | 59.76 | 89.15 | 70.88 |
| Alpaca | All | - | 70.66 | 72.31 | 70.42 | 96.50 | 71.48 |
| LLaMA | H&H | - | 73.00 | 72.53 | 64.02 | 90.81 | 72.76 |
| LLaMA | H&H | 0.1 | 72.10 | 72.23 | 59.29 | 87.74 | 72.17 |
| LLaMA | H&H | 1. | 71.21 | 71.75 | 57.77 | 90.33 | 71.48 |
| LLaMA | All | - | 73.08 | 72.62 | 71.34 | 96.84 | 72.85 |
| LLaMA | All | 1. | 73.17 | 72.53 | 70.27 | 96.53 | 72.85 |

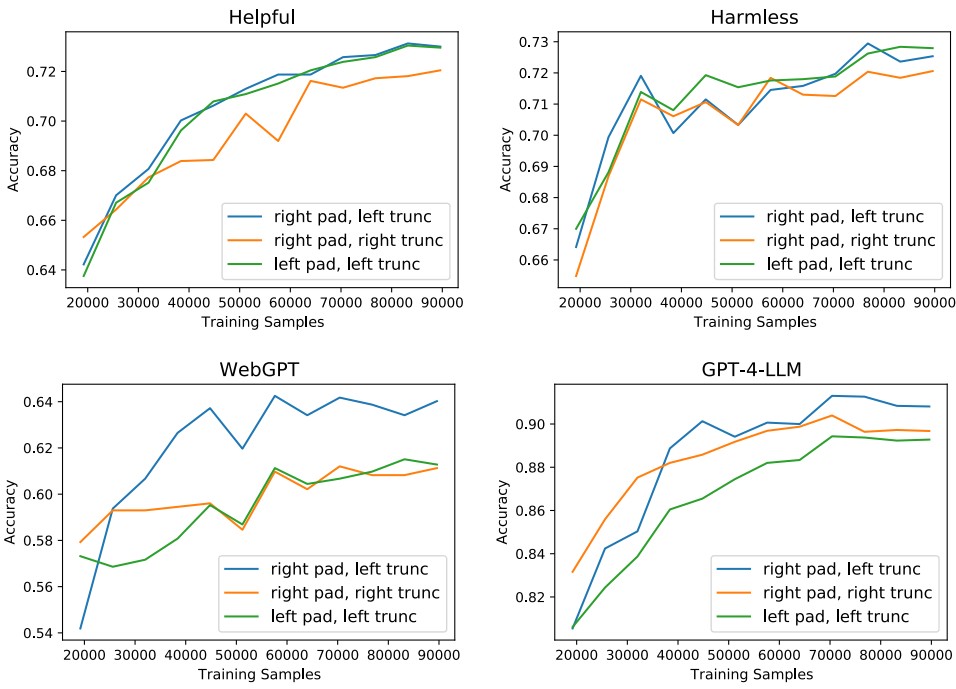

Figure 15: Ablation study of different input padding and truncation strategies on the GRFT stage.

Table 6: Testing performance of Alpaca-based RMs on the CRFT stage. "Alpaca" means the CRFT is conducted on the Alpaca base without the GRFT setup. "Alpaca + HH" means the CRFT is based on Alpaca with GRFT using H&H data.

| General RM | Domain | LM Coeff. | Helpful | Harmless | WebGPT | GPT-4-LLM | DSP |
|---|---|---|---|---|---|---|---|
| Alpaca | Business | - | 52.21 | 50.26 | 55.95 | 72.92 | 71.46 |
| Alpaca | Entertainment | - | 49.11 | 53.07 | 50.61 | 34.73 | 85.00 |
| Alpaca | Literature&Art | - | 52.76 | 48.83 | 53.81 | 52.60 | 79.88 |
| Alpaca | Normal | - | 44.39 | 52.68 | 38.87 | 24.71 | 90.20 |
| Alpaca | Academy | - | 55.91 | 46.93 | 54.12 | 81.79 | 75.30 |
| Alpaca + HH | Business | - | 67.01 | 72.32 | 57.01 | 88.90 | 73.81 |
| Alpaca + HH | Entertainment | - | 67.86 | 69.68 | 54.57 | 87.96 | 86.93 |
| Alpaca + HH | Literature&Art | - | 69.30 | 72.40 | 59.45 | 90.40 | 81.43 |
| Alpaca + HH | Normal | - | 64.07 | 71.58 | 39.02 | 55.64 | 93.00 |
| Alpaca + HH | Academy | - | 66.20 | 72.84 | 61.59 | 88.84 | 79.24 |
| Alpaca + HH | Business | 1. | 67.22 | 72.06 | 60.98 | 89.56 | 72.14 |
| Alpaca + HH | Entertainment | 1. | 68.07 | 67.52 | 58.69 | 90.04 | 84.74 |
| Alpaca + HH | Literature&Art | 1. | 69.09 | 68.60 | 64.18 | 92.89 | 77.04 |
| Alpaca + HH | Normal | 1. | 64.58 | 71.37 | 37.04 | 53.68 | 89.79 |
| Alpaca + HH | Academy | 1. | 67.43 | 71.24 | 65.55 | 89.85 | 77.27 |

### B.3 COMPARISON ON DATA SIZES AND IMITATION LEARNING

We report the final evaluation results of GRFT under different data and training setups in Table 5. From the results, we find LLaMA is uniformly better than Alpaca as the RM base under different training setups. With the same training data (H&H), enlarging the imitation learning coefficient can hinder the testing performance with both Alpaca and LLaMA bases. However, when enriching the training data samples, the impact of imitation learning on GRFT becomes smaller. Additionally, we show the performance comparison on the Alpaca base with different general training preferences (H&H, WebGPT, GPT-4-LLM). The general RM trained on WebGPT has the worst performance mainly because the training data size is too small ($\approx$ 20K), and the model is not well learned. However, the corresponding preference accuracy on GPT-4-LLM set is larger than 80%, which means the comparisons in the GPT-4-LLM set are easier.

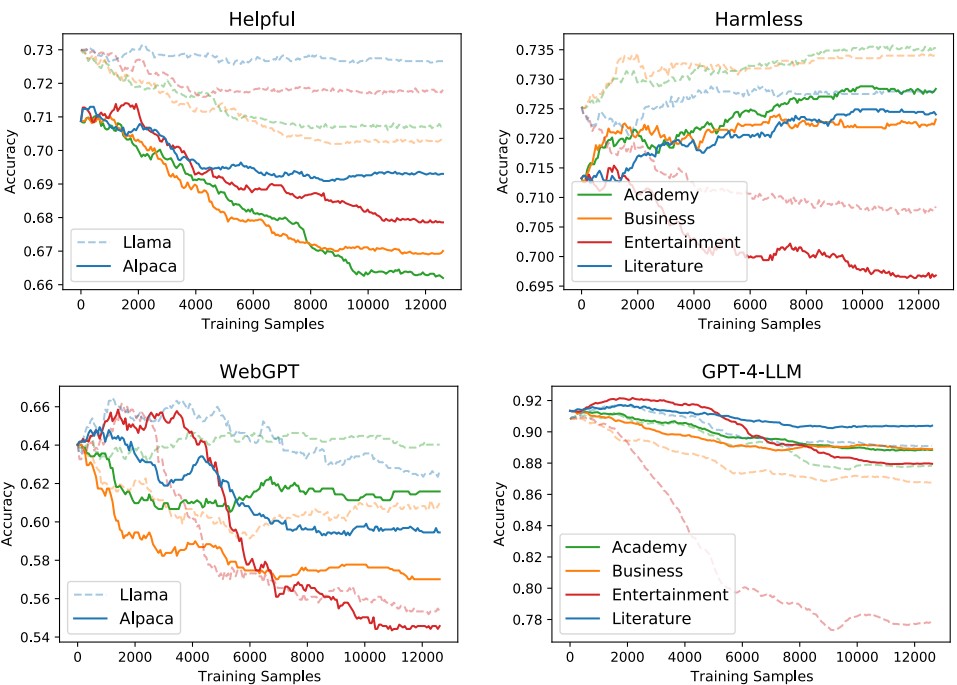

Figure 16: Additional testing performance on customized RM fine-tuning for base LM comparison.

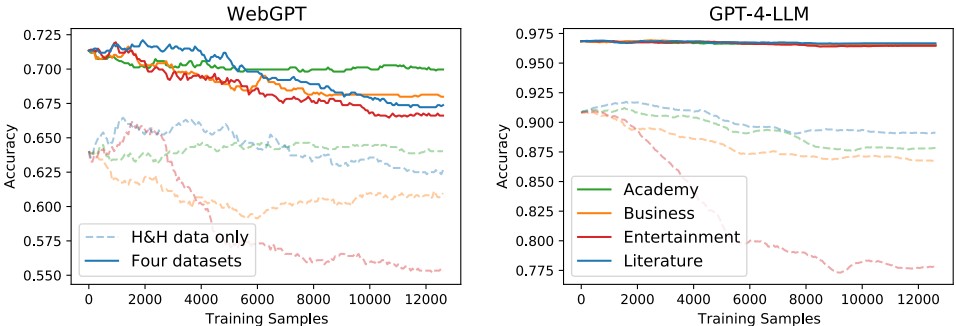

Figure 17: Additional CRFT testing performance with different GRFT data sizes.

## C   ADDITIONAL RESULTS OF CUSTOMIZED RM FINE-TUNING

### C.1   BASE MODEL COMPARISON

Besides the results in Figure 5, here we show the additional testing results of WebGPT and GPT-4-LLM in Figure 16. Although Alpaca-based RM still receives worse results on the WebGPT set, its performance on the GPT-4-LLM set is uniformly better. A possible reason for this phenomenon is that the data Alpaca used in SFT have the same instruction as the comparison data in GPT-4-LLM. Hence, the SFT data and the preference have a similar distribution with preference data, which provides a strong prior for the RM's generalization ability.

### C.2   TRAINING SAMPLE SIZES COMPARISON ON GENERAL RM FINE-TUNING

Besides the results in Figure 7, we show the additional testing results of WebGPT and GPT-4-LLM in Figure 17. There are clear performance gaps between LLaMA+All and LLaMA+H&H, obviously because LLaMA+All has already seen the train data of WebGPT and GPT-4-LLM in the GRFT stage. On both WebGPT and GPT-4-LLM sets, the general RM performance can be better preserved with the CRFT on *Academy* and *Literature&Art* domains than it on *Business* and *Entertainment*. This is probably because the prior preference datasets did not consider more diverse human preferences with the requirements in business activities and entertainment scenarios.

Table 7: Testing performance of Alpaca-based RMs on the CRFT stage. "LLaMA" means the CRFT is conducted on the LLaMA base without the GRFT setup. "LLaMA + HH" and "LLaMA + All" mean the CRFT is based on LLaMA with GRFT using H&H data and all data, respectively. "+LM" Means add imitation learning on the GRFT stage.

| General RM | Domain | LM Coeff. | Helpful | Harmless | WebGPT | GPT-4-LLM | DSP |
|---|---|---|---|---|---|---|---|
| LLaMA | Business | - | 53.23 | 54.80 | 52.44 | 68.29 | 72.33 |
| LLaMA | Entertainment | - | 52.72 | 46.45 | 53.51 | 41.67 | 85.27 |
| LLaMA | Literature&Art | - | 55.14 | 46.41 | 53.66 | 75.76 | 80.11 |
| LLaMA | Normal | - | 46.68 | 53.03 | 38.26 | 26.09 | 90.58 |
| LLaMA | Academy | - | 54.25 | 49.70 | 51.52 | 83.66 | 73.87 |
| LLaMA + HH | Business | - | 70.28 | 73.40 | 60.98 | 86.78 | **78.16** |
| LLaMA + HH | Entertainment | - | 71.73 | 70.80 | 55.34 | 77.77 | **88.29** |
| LLaMA + HH | Literature&Art | - | 72.66 | 72.79 | 62.35 | 89.11 | **84.87** |
| LLaMA + HH | Normal | - | 68.24 | 73.27 | 39.02 | 33.05 | **92.10** |
| LLaMA + HH | Academy | - | 70.71 | 73.53 | 64.02 | 87.82 | 80.94 |
| LLaMA + HH | Business | 0.1 | 70.28 | 73.27 | 60.67 | 87.11 | **78.16** |
| LLaMA + HH | Entertainment | 0.1 | 71.98 | 71.15 | 55.79 | 74.99 | 87.76 |
| LLaMA + HH | Literature&Art | 0.1 | 72.92 | 72.75 | 61.89 | 88.77 | 84.46 |
| LLaMA + HH | Normal | 0.1 | 68.20 | 73.27 | 39.63 | 30.12 | 91.68 |
| LLaMA + HH | Academy | 0.1 | 70.54 | **73.62** | 64.63 | 87.52 | 81.28 |
| LLaMA + HH | Business | 1. | 71.85 | 73.36 | 63.41 | 86.65 | 72.75 |
| LLaMA + HH | Entertainment | 1. | 71.39 | 71.54 | 53.66 | 78.62 | 86.10 |
| LLaMA + HH | Literature&Art | 1. | **73.38** | 72.53 | 64.79 | 90.28 | 78.44 |
| LLaMA + HH | Normal | 1. | 67.86 | 73.31 | 36.28 | 23.69 | 90.32 |
| LLaMA + HH | Academy | 1. | 71.90 | 73.05 | 64.48 | 83.57 | 78.48 |
| LLaMA + All | Business | - | 71.30 | 73.53 | 67.99 | 96.50 | 76.00 |
| LLaMA + All | Entertainment | - | 71.90 | 72.15 | 66.62 | 96.46 | 87.61 |
| LLaMA + All | Literature&Art | - | 72.19 | 73.75 | 67.38 | 96.67 | 83.62 |
| LLaMA + All | Normal | - | 69.81 | 73.27 | 43.90 | 78.50 | 91.98 |
| LLaMA + All | Academy | - | 70.83 | 73.44 | **69.97** | 96.52 | 81.39 |
| LLaMA + All | Business | 0.1 | 71.26 | 73.36 | 68.90 | 96.37 | 76.04 |
| LLaMA + All | Entertainment | 0.1 | 72.11 | 72.15 | 65.24 | 96.35 | 87.69 |
| LLaMA + All | Literature&Art | 0.1 | 72.15 | 73.79 | 67.23 | **96.69** | 83.32 |
| LLaMA + All | Normal | 0.1 | 69.47 | 73.27 | 42.99 | 76.63 | 91.68 |
| LLaMA + All | Academy | 0.1 | 70.88 | **73.62** | 69.36 | 96.37 | **81.47** |
| LLaMA + HH + LM(1.) | Business | - | 69.39 | 71.76 | 57.62 | 75.91 | 75.55 |
| LLaMA + HH + LM(1.) | Entertainment | - | 71.00 | 69.29 | 54.88 | 57.36 | 87.27 |
| LLaMA + HH + LM(1.) | Literature&Art | - | 70.66 | 71.54 | 55.18 | 78.96 | 83.36 |
| LLaMA + HH + LM(1.) | Normal | - | 68.79 | 70.80 | 39.33 | 16.41 | 90.85 |
| LLaMA + HH + LM(1.) | Academy | - | 69.90 | 71.76 | 60.98 | 75.97 | 80.56 |
| LLaMA + HH + LM(0.1) | Business | - | 71.05 | 73.18 | 58.08 | 74.63 | 75.66 |
| LLaMA + HH + LM(0.1) | Entertainment | - | 72.19 | 70.20 | 52.74 | 74.97 | 86.86 |
| LLaMA + HH + LM(0.1) | Literature&Art | - | 71.43 | 72.97 | 60.06 | 82.59 | 83.96 |
| LLaMA + HH + LM(0.1) | Normal | - | 69.90 | 72.97 | 41.16 | 23.18 | 91.60 |
| LLaMA + HH + LM(0.1) | Academy | - | 70.03 | 73.14 | 62.96 | 78.14 | 81.32 |

## C.3 IMITATION LEARNING ON GENERAL FINE-TUNING

In Figure 18, we show the performance changes on the four testing sets during CRFT. With imitation learning in the GRFT stage, there is no uniform accuracy improvement. The performance on the GPT-4-LLM set gets even worse. This is probably because the text data distributions of H&H and GPT-4-LLM are quite different, where the language modeling loss on H&H pushes the general RM away from the GPT-4-LLM data distribution.

## C.4 IMITATION LEARNING ON CUSTOMIZED FINE-TUNING

In Figure 19, we show the CRFT testing performance with imitation learning coefficient $\mu = 0.1$. With imitation learning, the performance on the GPT-4-LLM set is uniformly getting worse. However, the performance on *Helpful* and *Harmless* sets can generally be better preserved compared to the fine-tuning baselines.

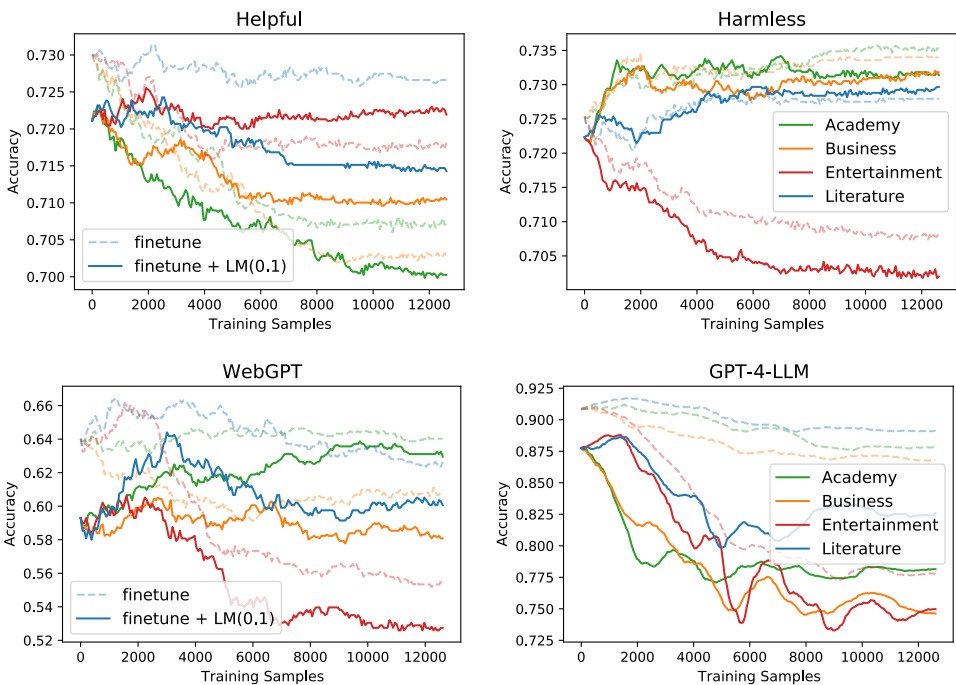

Figure 18: Additional testing performance on customized RM fine-tuning with GRFT adding imitation learning loss.

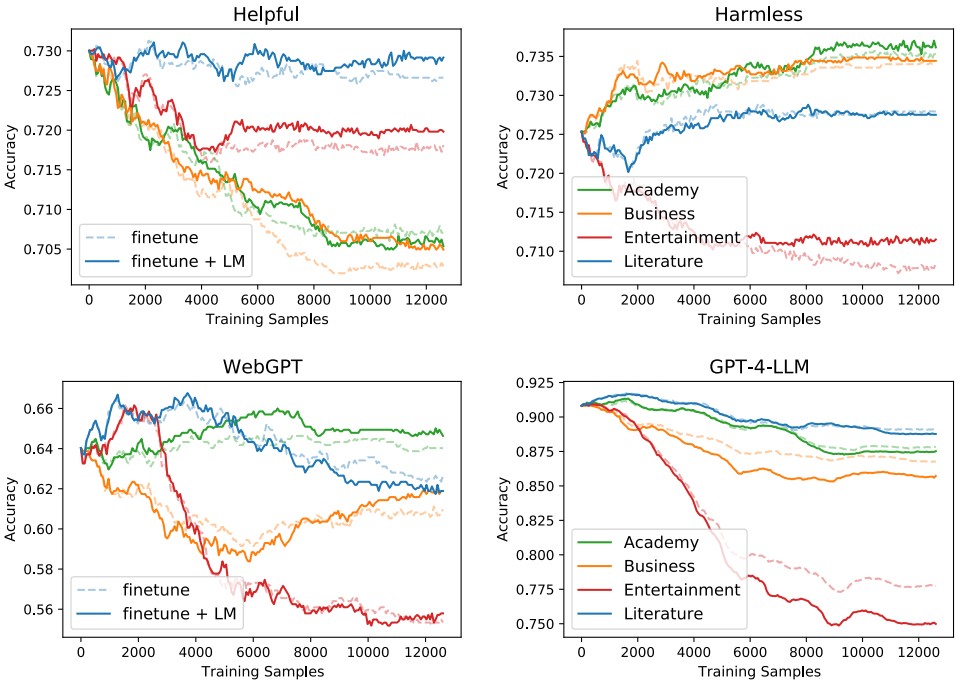

Figure 19: Impact of language modeling loss in domain-specific RM fine-tuning.

## C.5 ABLATION STUDY OF IMITATION LEARNING COEFFICIENTS ON CRFT

In Figure 20, Figure 21, and Figure 22, we show the ablation study of the LM coefficients on domains *Entertainment*, *Aacademy*, and *Literature&Art*, respectively. Besides the plot on the *Business*

domain, the other three plots also verify our discovery that the smaller LM coefficient ($\mu = 0.1$) can preserve the general RM performance with almost no loss on the customized RM ability. Moreover, on the *Entertainment* domain, imitation learning with the smaller coefficient ($\mu = 0.1$) even reaches a higher general RM performance than it with the larger coefficient ($\mu = 1.$). On the *Academy* domain, CRFT with imitation learning ($\mu = 0.1$) even achieves a higher accuracy than the baseline customized RM learned with the ranking loss.

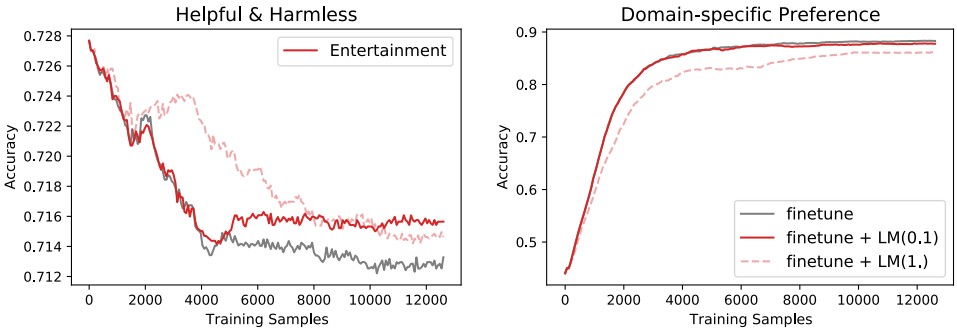

Figure 20: Ablation study of LM loss coefficient on *Entertainment* Domain.

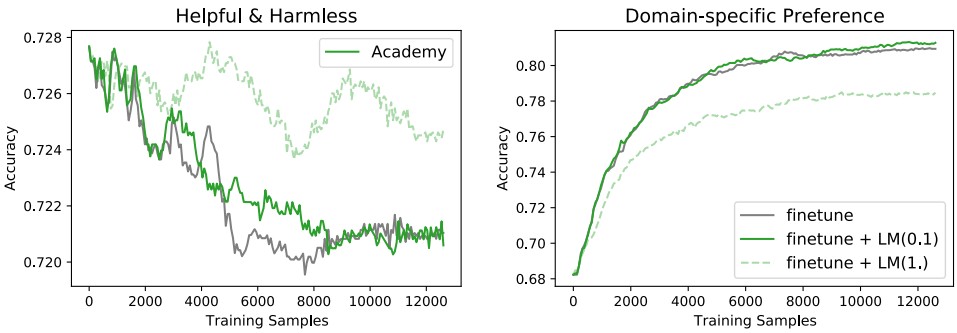

Figure 21: Ablation study of LM loss coefficient on *Academy* Domain.

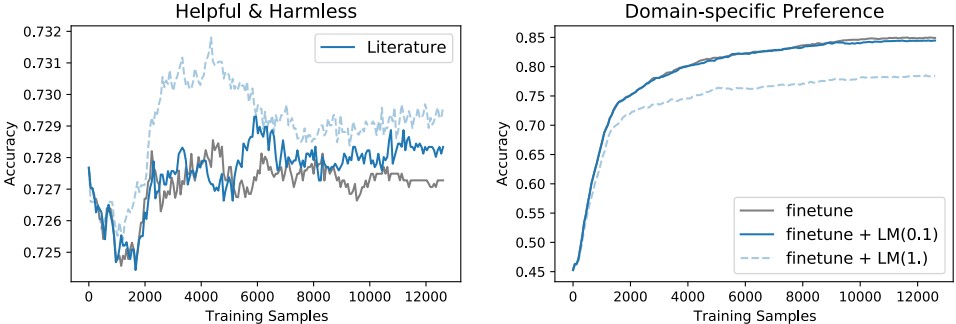

Figure 22: Ablation study of imitation coefficient on CRFT of the *Literature&Art* Domain.

# D   SAMPLES IN THE DSP DATASET

| | |
|---|---|
| Query | Generate a story about a musician in less than 40 words. |
| Normal | Gina was a talented musician with a passion for music. She would spend hours perfecting her craft until she was nearly perfect. She eventually became a successful and celebrated artist and her concerts were always sold out. |
| Academy | After years of playing local gigs, the musician finally got their big break when a record label executive stumbled upon their acoustic set and offered them a contract. |
| Business | "Born with a natural talent for music, she spent countless hours honing her skills. Soon, the world recognized her as a true artist - a musician who had the power to move people with her melodies." |
| Entertainment | Once a struggling musician, she finally got her big break when her song went viral. She now performs sold-out shows worldwide, inspiring others to follow their dreams. |
| Literature&Art | His fingers danced across the guitar strings, pouring his heart out in every note. The audience was spellbound, swaying to the rhythm, feeling every emotion he conveyed through his music. |
| Query | Generate a headline for the event:
"Introducing the 2020 ARR Scholarship Program". |
| Normal | New Scholarship Opportunity for 2020:
ARR Launches their 2020 Scholarship Program! |
| Academy | "Apply now for the 2020 ARR Scholarship Program and boost your academic career". |
| Business | "Unlocking Opportunities:
Announcing the Launch of the 2020 ARR Scholarship Program" |
| Entertainment | "Get Money to Get Smart:
The ARR Scholarship Program is Back and Better Than Ever for 2020!" |
| Literature&Art | "Opening Doors to Education:
2020 ARR Scholarship Program Now Accepting Applications" |
| Query | Describe the economic system of the United States. |
| Normal | The United States operates under a mixed economy, which is a system that combines free-market capitalism with governmental, economic regulations. This system allows businesses to compete in a free, competitive market while protecting consumers and providing employment and safety regulations. Additionally, the government provides social services, including education, health care and infrastructure. In this way, it seeks to balance the needs of both the market and society. |
| Academy | The economic system of the United States is primarily capitalist, characterized by individual economic freedom and relatively minimal government intervention. The government provides a legal and regulatory framework, but for the most part, businesses are free to operate and compete in the marketplace. The United States has a mixed economy, as there are some government-run enterprises and a degree of social welfare programs, but it is primarily driven by the private sector. The country's economy is diversified, including a mix of agricultural, manufacturing, and service sectors. The currency used is the US Dollar and the country is a member of the World Trade Organization and other international economic organizations. |

| Business | The United States has a mixed economic system which combines elements of both capitalism and socialism. The government plays a significant role in regulating the economy and providing public services, but also allows for private enterprise and competition. The country has a free-market economy which means that individuals and private firms can engage in economic activities without significant interference by the government. However, the government does intervene in certain areas such as banking, agriculture, education, and healthcare. In addition to private enterprise, the United States also has public services such as social security, Medicare, and Medicaid, which provide safety nets for citizens in need. Overall, the economic system in the United States is dynamic and constantly evolving, driven by market forces, technological innovations, and government policies. |
|---|---|
| Entertainment | Well folks, let's talk about the economic system of the land of the free, home of the brave. The United States has a mixed economy, which means it combines elements of capitalism and socialism.
On one hand, the United States is known for its free market economy, where businesses and individuals are free to buy and sell goods and services with limited government intervention. This has led to a lot of innovation and growth in industries like technology and entertainment.
On the other hand, the United States also has a number of social programs and safety nets, like Social Security and Medicare, which are funded through taxes and provide benefits to Americans.
In short, the economic system of the United States is like a sandwich - a mix of different ingredients, but hopefully still tastes good in the end. |
| Literature&Art | Ah, a question of economics, a subject both complex and fascinating. To speak of the United States' economic system is to delve into a world of trade, markets, and wealth.
At its core, the US economy functions as a mixed-market system. This means that while the government plays a role in regulating business and controlling certain aspects of the economy, private enterprise is allowed to operate in a free market, with supply and demand driving prices and production.
The United States is also characterized by its vast and diverse economy, made up of different industries and sectors, including manufacturing, finance, agriculture, and technology. This diversity has allowed for innovation and growth, creating a thriving economy that has had a powerful impact on the global stage.
Of course, no system is without its flaws. The United States has faced challenges such as income inequality, as well as periods of economic recession and depression. But with ingenuity and perseverance, the American people have always found a way to weather such storms and emerge stronger on the other side.
In summary, the United States' economic system is a complex and ever-evolving mix of private enterprise and government regulation, fueled by a diverse array of industries and characterized by a drive for innovation and growth. |
| Query | Come up with a 10-word sentence that begins with "I love". |
| Normal | I love the smell of freshly cut grass in the summertime. |
| Academy | I love spending time exploring new places with my friends. |
| Business | I love spending time with my family and close friends. |
| Entertainment | I love laughter, especially when it's at my own expense. |
| Literature&Art | I love the poetry of raindrops on my window pane. |
| Query | Explain two effects of climate change on the environment |

| | |
|---|---|
| Normal | Climate change has several effects on the environment. One effect is an increase in global average temperature, which leads to melting of icebergs, extreme weather events, and a rise in sea levels. Another effect is a decrease in biodiversity due to the destruction of natural habitats caused by extreme weather conditions and rising temperature. |
| Academy | Climate change has significant impacts on the environment, and some of the most notable effects include:
1. Melting of Glaciers and Ice Caps: One of the most visible effects of climate change is the melting of glaciers and ice caps. As the global temperature continues to rise, the amount of ice in the polar regions and on mountains decreases, leading to rising sea levels and changes in ocean currents. Besides, the melting ice also disturbs the ecosystem of the Arctic and Antarctic regions, affecting the survival of several animals that are adapted to the polar conditions.
2. Changes in Ecosystems: Climate change leads to changes in the range and distribution of plant and animal species. As the temperature, rainfall, and humidity levels change, plants and animals move to adapt to new conditions leading to new patterns of biodiversity. This displacement of species also affects the food chains and habitats of many ecosystems, leading to a decline in biodiversity. Furthermore, the changes in ecosystems can also impact human well-being as they can affect the availability of resources, such as water and food. |
| Business | Climate change has numerous effects on the environment, but two of the most significant effects are:
1. Melting of Glaciers and Rising Sea Levels: Climate change has caused the temperature of the earth to rise, which has led to the melting of glaciers and polar ice caps. This has resulted in an increase in sea levels, which has made some islands and coastal areas at risk of flooding. The increase in sea levels has also caused the loss of habitats for marine animals and birds that rely on coastal ecosystems.
2. Changes in Weather Patterns: Climate change has altered weather patterns, leading to more severe weather conditions like floods, droughts, hurricanes, and storms. These conditions have impacted the ecosystems, as prolonged droughts can lead to forest fires and a shortage of resources like water and food. Additionally, severe weather patterns have led to the extinction of several animal and plant species, as they are unable to adapt to the changing climate. |
| Entertainment | Sure thing! Climate change has a number of devastating effects on our environment, but let me highlight two of the most prominent:
1. Rising temperatures: As global temperatures rise, we are seeing more frequent and severe heatwaves. This can lead to droughts and wildfires in some areas, while other regions may experience more powerful storms and flooding. Additionally, increasing temperatures can lead to the melting of glaciers and ice caps, which causes sea levels to rise. This can threaten coastal communities and habitats, and can even lead to the displacement of people who live in those areas.
2. Ocean acidification: The oceans absorb a lot of the carbon dioxide that we produce, which has a knock-on effect of increasing the acidity of seawater. This can have a catastrophic impact on marine life, particularly on species with hard shells or skeletons. The acidity of the water can damage or dissolve shells and other structures, which can kill off species that are essential to the ocean ecosystem. This can have a domino effect, leading to other species being impacted as well, and can ultimately undermine the entire food chain. |

| Literature&Art | As a poet, let me describe the effects of climate change on the environment in verse:
The planet's temperature rises, and ice caps melt,
The oceans warm and acidic, as carbon dioxide is felt.
The weather becomes unpredictable, with droughts and floods,
Habitats shift and disrupt, as wildlife face new struggles.
As the planet heats up, the environment is at stake,
Glaciers thaw, seas rise, and ecosystems break.
Extreme weather, changing seasons, and storms that rage,
All cause damage, as the climate turns a new page.
The Earth's beauty is at risk, as we neglect our role,
It's time to act, and make changes, to save each living soul.
For the sake of our planet's future, let's work together,
To heal the world, and make it a sustainable endeavor. |
|---|---|
| Query | Complete the following analogy: "A relationship is like a ______ because ______." |
| Normal | A relationship is like a garden because it requires dedication, care, and patience to tend to it and help it grow. |
| Academy | A relationship is like a plant because it requires continuous care and nourishment in order to grow and thrive. Without proper attention and effort, it may wither and die. |
| Business | A relationship is like a plant because it requires consistent nurturing, attention, and care to grow and thrive. Lack of attention and care can result in the relationship wilting or dying. |
| Entertainment | A relationship is like a rollercoaster because it has its ups and downs, thrilling moments, and can make you feel a little bit sick at times, but ultimately leaves you wanting more. |
| Literature&Art | A relationship is like a garden because it requires constant care, attention, and nurturing to flourish and grow. Just like a garden needs watering, pruning, and weeding to thrive, a relationship needs communication, compromise, and understanding to withstand the challenges that come its way. Just as a garden can bring beauty and sustenance, a strong and healthy relationship can bring joy, support, and fulfillment in our lives. |
| Query | Generate an analogy comparing a cat to a computer. |
| Academy | A cat could be compared to a computer in terms of their respective modes of operation. Just as a cat can be inquisitive, curious and unpredictable, a computer can effectively operate in a manner that is adaptable, efficient and precise. Similarly, while a cat may occasionally have a "glitch" in their behavior or unexpected response to a stimulus, the same can be true of a computer with programming errors or hardware issues. Ultimately, both a cat and a computer have the potential to be highly effective creatures when properly managed and maintained. |
| Business | A cat can be compared to a computer in the sense that both have a certain level of independence and can operate on their own, without constant supervision. Just as a cat can roam around the house and entertain itself, a computer can run automated programs and perform tasks without human intervention. However, both a cat and a computer need attention and maintenance to ensure they function properly. Just as a cat needs food, water, and occasional grooming, a computer needs regular updates and virus scans to maintain its performance. |
| Entertainment | Well, let me think... Ah! Here's one: cats are like computers. Sometimes they work perfectly and you're really happy to have them around, but other times they just stop cooperating and leave you frustrated and confused. And just like with computers, you can try rebooting them or giving them a little tap, but ultimately you're at their mercy. But hey, we still love them anyway, don't we? |

| Literature&Art | A cat is like a computer in the way that it processes information, runs countless internal programs, and occasionally crashes when overloaded with too much data. Both require proper maintenance and occasional updates to continue functioning at optimal levels. And just as a computer can sometimes seem inscrutable to the uninitiated, so too can a cat's thought processes and behaviors appear mysterious and difficult to comprehend. |
|---|---|
| Query | Describe an armadillo in three sentences. |
| Academy | An armadillo is a small mammal native to the Americas. It has a hard, protective shell covering its body, and is often used as an example of an animal with armor. Despite their armored defense, armadillos are relatively slow-moving and often preyed upon by larger predators. |
| Business | The armadillo is a mammal with a protective shell made of bony plates. It is native to the Americas and is known for its unique digging ability. Armadillos are omnivores and primarily feed on insects, small animals and plant matter. |
| Entertainment | Well, an armadillo is like a walking tank, covered in armor-like scales that protect it from predators. These strange little creatures are also famous for their ability to curl up into a ball at the slightest sign of danger, making them quite the tough target to take on. And, believe it or not, armadillos are actually skilled diggers that use their strong legs and sharp claws to root around in the dirt for food. |
| Literature&Art | The armored armadillo, a creature unique, with plates of shells that no other beast can critique. A ball it can roll up into with ease, its armor so strong, you cannot make it sneeze. Strange and fascinating, it roams the forest floor at night, an animal with a charm that's truly out of sight. |