# OpenReview forum: "Everyone Deserves A Reward: Learning Customized Human Preferences"
_ICLR.cc/2024/Conference — Submitted to ICLR 2024_

### Official Review · Reviewer_NCvM · 2023-11-01

**Soundness:** 3 good
**Presentation:** 3 good
**Contribution:** 2 fair
**Rating:** 5
**Confidence:** 2

**Summary:**

The paper addresses the limitations of current human feedback aligning methods which typically use a general reward model, failing to satisfy diverse, customized preferences. To address the issue, the authors created a DSP dataset consisting of user queries and corresponding responses from four domains. This dataset was used to train both general and domain-specific reward models. A three-stage training scheme is proposed, which involves a base language model training, a general RM fine-tuning, and customized RM fine-tuning. Multiple training and data strategies were tested across these stages to find ways to fit customized preferences while preserving the general preference capability of the models.

**Strengths:**

- The generated DSP dataset can be useful to the community.
- The three-stage training scheme for customized RM learning looks legit to me.
- The discovery that imitation learning on customized preferences and general preference data enrichment preserves the RMs’ general preferring ability when fitting customized human preferences is interesting.

**Weaknesses:**

It seems to me that the DSP dataset could inherently contain biases based on the chosen domains. How these biases are identified and mitigated is not clearly addressed, which is crucial in a study aiming to cater to diverse human preferences. The efficacy of the training scheme is tested on a specific dataset (DSP). The extent to which these findings can be generalized to other datasets or real-world scenarios is also not very clear.

**Questions:**

- How can we ensure the diversity and representativeness of the DSP dataset?
- Can you highlight the technical contribution of the proposed three-stage training scheme?

---

### Official Review · Reviewer_AsgA · 2023-11-01

**Soundness:** 2 fair
**Presentation:** 2 fair
**Contribution:** 2 fair
**Rating:** 3
**Confidence:** 3

**Summary:**

The paper presents a new way of generating fine tuning examples that are tailored to a specific context (business, entertainment, etc.)
hence are more adapted and fine tuned to a particular user need. The generated answers are scored by another model given the relevant context. These fine tuning examples are then used to train the model with a ranking and imitation loss function. The authors also claim to learn customized human preferences although it is somewhat unclear how this is achieved as there is  a very compressed description of the process. The resulting models are then evaluated but again it is unclear to me how to interpret these results as the baselines are mostly absent. Moreover there is no comparison to other state-of-the-art fine-tuning techniques.

**Strengths:**

Interesting approach in generating context depended samples

Potentially interesting simple fine-tuning techniques

**Weaknesses:**

The paper could benefit from more detailed explanations of the data generation process and the used in the approach.
It does not provide a comparison of their approach with other state-of-the-art methods in the field.
Overall the clarity of the paper could be improved

**Questions:**

Clarity needs to improve.

The contribution of the paper seems a bit weak, is the context-depended sample generation the main point, is this leading to a more personalized model ? I could not see this in the experiments.
How does this work compare to the state-of-the art in the area?

---

### Official Review · Reviewer_kam2 · 2023-11-10

**Soundness:** 4 excellent
**Presentation:** 3 good
**Contribution:** 2 fair
**Rating:** 5
**Confidence:** 4

**Summary:**

This paper addresses the challenge of developing personalized reward models for language model alignment. It highlights the importance of customizing reward models to cater to individual preferences or specific use-cases. The main contributions are the introduction of a novel synthetic preference dataset specifically designed for the customization of reward models, and a baseline training methodology employing multi-stage fine-tuning.

The paper explores the impacts of different configurations of the baseline training methodology on the performance of these customized reward models. This includes examining the role of an intermediate fine-tuning step with general preference datasets in enhancing final performance as well as the utilization of imitation-learning regulation in different stages of the training process. The latter aims to balance the fine-tuning process on smaller, personalized datasets, mitigating overfitting and maintaining generalization capabilities. Notably, the study indicates potential drawbacks of excessive fine-tuning steps, as can be seen by the relative performances of the Alpaca and LLaMA-based models.
Empirical results are presented to validate the proposed training techniques, offering insights into the feasibility and effectiveness of the approach.

I enjoyed reading the paper and appreciate the direction of this research. I encourage the authors to continue to pursue this direction, address the current weaknesses and possibly extend their work for an even stronger submission in the future.

**Strengths:**

- **Originality**: The paper proposes a novel benchmark dataset that can foster work into the direction of reward-model customization and  for individual user preferences or specific use cases. They additionally propose a baseline approach to tackle this dataset, which is of limited novelty.

- **Quality**: The methodology presented in the paper, particularly the development of a new synthetic dataset and a multi-stage fine-tuning training process, demonstrates a high level of technical soundness. The empirical evaluation is thorough, effectively balancing the fine-tuning performance with the need to maintain generalization and avoid overfitting to specific custom preferences.

- **Clarity**: The paper is well-articulated, offering a clear explanation of the concepts and methodologies employed. The use of empirical data to support claims and the thorough referencing of relevant literature in the field of language-model alignment add to the paper's clarity and credibility.

- **Significance**: The introduction of a dataset specifically for reward model customization is of significant value, as it provides a benchmark for future research in this area. It highlights an important problem and may motivate further work in this direction. The insights into the balance between fine-tuning for personalization and maintaining generalizability are valuable for the ongoing development of customized language models.

**Weaknesses:**

### Primary concerns:

- Limited Novelty of Baseline Approach: While the introduction of the benchmark dataset is innovative, the baseline approach proposed for addressing this dataset is somewhat derivative, primarily building upon the work by Askell et al. (2021). This is not in itself problematic if we consider the dataset to be the main contribution of the paper, but combined with the synthetic nature of the dataset it limits the significance of the contribution.

- Synthetic Nature of Dataset: The use of a synthetic dataset, generated by a language model, raises concerns about its practical utility. The dataset might not adequately represent real-world complexities of individual preferences, as it is model-generated. In particular, the fact that a language model trained on general human preferences was able to generate these personas calls the need for customized reward models into question for this particular dataset. Addressing this limitation by incorporating or comparing with real human-generated data would significantly strengthen the paper.

- Lack of Baselines and Alternatives: Since (as mentioned in the previous point) prompted language models already show significant capability of customizing their outputs to align with the preferences described in their prompt, comparison of fine-tuned reward models to a prompted language model would further strengthen the contribution.

- Lack of Policy-Level Evaluation: The paper focuses exclusively on reward-model level evaluation. Including policy-level evaluations, such as example outputs of language models fine-tuned with the customized reward models, would provide a more comprehensive understanding of the practical applications and effectiveness of the approach.

- Insufficient Discussion on Personalization: While the background on language-model specific alignment techniques is quite solid, the paper does not sufficiently explore prior work in personalization within the realms of general preference learning and information retrieval. Expanding the discussion to include these fields could provide a richer context and potentially lead to new directions to explore.

### Secondary concerns:

- Structural Improvements: The current structure of the paper could be optimized for better readability and impact. The paper currently puts large emphasis on the evaluation of the imitation learning generalization, although this is not the main contribution in my view. It could be improved by presenting the most promising results (Figure 11) first and only then discussing the ablations.

- Clarity in Abstract and Terminology: The abstract could be a bit clearer. On first read, I did not understand what you mean by "data strategies", and "training strategies" is similarly vague. I suggest to be more explicit, i.e., explain that you test your method with varying combinations of datasets in the intermediate fine-tuning stage and that you evaluate the impact of "imitation regularization". The abstract could additionally explain the relation between individuals and domains better. As it is, it is slightly confusing that you first call for the need for individualized reward models but then propose a domain-specific (rather than individual-specific) dataset.

### Minor points that have not impacted the score:

- The sentence "We discovered that imitation learning on customized preferences and general preference data enrichment are the two effective ways to preserve RMs’ general preferring ability when fitting the customized human preferences." could use clarification. While I understand it after reading your entire paper, its meaning was not clear to me in the beginning. It would help to clarify what you mean by "imitation learning on customized preferences" (i.e., adding the imitation loss in addition to the comparison loss) and "general preferring ability".

- The terms "language modeling" and "LM coefficient" are not quite self-explanatory, since the entire task is in a sense about language modeling. You mix those terms with "imitation learning" and "imitation learning loss", which I think cover the concept better.

- The order of the figures and tables is sometimes confusing since it does not match the order in which they are discussed in the text. I think reordering them would improve the reading experience. An example of this are Figure 5 and 6, which should be swapped.

- The conclusion of 4.4 ("Although facilitated the general preference preservation, the imitation learning results on the GRFT stage are not satisfying enough for CRFT.") only becomes clear in light of the later results with imitation-learning regularization for CRFT. It would help to rephrase that sentence.

- While the paper is generally well written and articulated, there are some terms and phrases that struck me as a bit awkward and could benefit from rephrasing. Among them are "pretraining with tremendous tokens", "guide the aligning directions", "helpfulness and harmlessness cover a wide range of mankind's tendencies", "general preference ability" / "preferring ability" and "costs a mess of annotation resources" and "let ChatGPT play as an assistant".

**Questions:**

1. Could you describe your experimental setup in some more detail? What exactly does "training samples" on the X-axis on all the plots refer to?

2. The motivation section suggests that high-quality customized reward models can enhance domain-specific LLM fine-tuning. However, reward-driven fine-tuning typically refines existing capabilities rather than adding new knowledge. Could you elaborate on how customized reward functions could enable language models to effectively handle novel application domains that LLMs fine-tuned with general preferences struggle with?

3. You mention that the three-stage training scheme for customized RM training is one of your main contributions, yet acknowledge its similarity to the scheme proposed by Askell et al. (2021). Could you clarify if and how your training scheme differs from theirs?

4. You note that collecting customized preferences from different persons could make the labeling task more difficult than gathering general preferences. Can you elaborate on why this is the case?

5. Regarding the selection of instructions from the Alpaca dataset, you mention no requirement on the "input" key. Could you clarify what this means?

6. In Figure 8, the term "Llama Base" is used. Could you clarify what this refers to? Is it "Llama + LM(0.0)"? Additionally, on which datasets are the models in this figure trained?

---

> ### Author Response · Authors · 2023-11-23
>
> We sincerely appreciate Reviewer Kam2 for such an elaborate review, which provides us with a lot of inspiration and suggestions. We will take a hard look at the reviewer's concerns and further polish this work to be better.
>
> Response to the reviewer's primary concerns:
> 1. Novelty of Approach: we agree that the imitation learning method we used in GRFT and CRFT steps is similar to the PMP method in Askell et al.(2021). However, we use this imitation learning loss for a different purpose: PMP uses imitation learning to improve the performance of a general RM; we use imitation learning to preserve the general RM performance when learning a customized RM. Our main contribution to the methodology part is we find this imitation learning loss can enhance a new task CRFT, which is practically valued for many LLM customization scenarios.
>
> 2. Synthetic Nature of Dataset: We agree that the dataset we release is somehow a synthetic dataset, at least not annotated by humans. We will extend this dataset with more complicated preferences and more large size. However, the main point of the DSP dataset is to quickly build scenarios for customized RM training and to quickly verify the effectiveness of DSP baseline approaches.
>
> 3. Lack of Baselines: The reviewer provides a good suggestion to use the prompted LLM for evaluation as a baseline. We will add this baseline in the future revision. However, the performance of prompted LLM can be with high variance for the variety in prompt selection.
>
> 4. Lack of Policy-Level Evaluation: Although we pay most of our attention to the RM learning task, we acknowledge that adding policy-level evaluation can further verify the effectiveness of the three-stage RM training scheme. However, evaluating domain-specific LLM can be far more challenging: we need annotators to act as a person from a particular domain, which requires additional annotation resources.
>
> 5. Discussion on Personalization: We will add more personalization-related introductions in the future review. Thanks for the suggestion.
>
> Response to the reviewer's secondary concerns:
>
> 1. Structural Improvements: we will reorder the experiments and highlight our discovery of imitation learning in the next revision.
>
> 2. Clarity in Abstract and Terminology: We will make the abstract more clear. Our initial logic is: both domain-specific and individual-specific preferences are customized preferences. However individual-specific preferences are difficult to collect so we collect domain-specific preferences instead.
>
> For the minor points, we appreciate the reviewer's careful check and will revise them in the draft.
>
> Response to the reviewer's questions:
>
> 1. "training samples" means the number of comparison pairs.
>
> 2. Similar to primary concern 4, we will try to add policy-level experiments. However, the evaluation of domain-specific LLM remains challenging.
>
> 3. The major difference between our three-stage training scheme and PMP is the training purpose: our training scheme is used for customized preference learning while preserving general preference performance, but PMP is used for general preferences. Besides, PMP uses far larger pretraining sets, while our scheme uses the PMP finetuning data on the GRFT step.
>
> 4. When considering each person, the preference collection situation can be much more complicated. Because people can have agreement on some topics while having divergence on others. Moreover, the degree of divergence can be different depending on the person. It is challenging to collect preferences from a few annotators to reflect the variety of divergence to an appropriate degree.
>
> 5. The queries in the Alpaca dataset contain two keywords: "instruction" and "input". While all queries include "instruction", some of them do not include "input". We simply exclude queries with "input" for simplification.
>
> 6. Yes, "Llama Base" equals "Llama + LM(0.0)". The model is trained on the original HH dataset as for GRFT.

---

### Meta-Review · Area_Chair_FnSS · 2023-12-09

**Metareview:**

The authors address the task of developing personalized reward models for language model alignment, highlighting the importance of customized reward models tailored to individual user preferences. The main contributions are the introduction of a novel synthetic preference dataset specifically designed for the customization of reward models, and a baseline training methodology employing multi-stage fine-tuning. The authors also explore the impact of different configurations of the baseline training methodology on the performance of these customized reward models.

Although the reviewers agree that the paper is interesting and makes a step in the right direction, they also raise a number of critical comments and concerns, including the limited novelty of the baseline approach, the synthetic nature and possible bias of the dataset, the lack of baselines and alternatives, the lack of policy-level evaluation, the insufficient discussion on personalisation, and questions regarding generalisation. In the discussion between authors and reviewers, some of these points could be resolved but other not. Eventually, there was a consensus that the paper does not yet reach the level expected for a top-venue such as ICLR.

**Justification For Why Not Higher Score:**

Quality is not good enough.

**Justification For Why Not Lower Score:**

N/A

---

### Decision · Program_Chairs · 2024-01-16

Reject